# Does it hold weight? The perceived effects of contraceptive use on weight status in females: A mixed-methods study

Leonie Bass☯, Tamara Prostináková☯, Kathelijne Gabrielle Silang☯,
Alyssia Griffiths-Gray☯, Stephen McQuilliam☯, Elizabeth Mahon☯,
Amy Whitehead☯, Kelsie Olivia Johnson☯*

Research Institute for Sport and Exercise Science, Liverpool John Moores University, Liverpool, United Kingdom

☯ These authors contributed equally to this work.

* k.o.johnson@ljmu.ac.uk

## Abstract

### Purpose

To examine females' perceived effects of contraceptive use on weight management and to identify potential confounding factors that may contribute to perceived weight changes across their lifespan.

### Methods

Three hundred and fifteen predominantly UK-based females completed a questionnaire assessing the prevalence and type of current and previous contraceptive use, reasons for use, side effects experienced, and perceived effects on weight. Twenty-five participants who reported contraceptive use subsequently completed timeline interviews to gain deeper insight into the perceived impact of contraception on weight status and to identify contextual or lifestyle factors that may have contributed to perceived weight-management difficulties.

### Results

Contraceptive use prevalence was 88% among respondents, with 38% using a contraceptive method at the time of data collection. Forty-two percent reported greater difficulty managing their weight while using hormonal contraception compared with their natural menstrual cycle. In follow-up interviews, *depot medroxyprogesterone acetate (DMPA)* was consistently identified as the method most associated with difficulties in weight management. However, most participants attributed weight-related challenges to broader life circumstances, such as moving away from home, relationship changes, and stress, rather than contraceptive use alone.

**Data availability statement:** All relevant data are within the paper and its Supporting Information files.

**Funding:** The author(s) received no specific funding for this work.

**Competing interests:** The authors have declared that no competing interests exist.

## Conclusion

Although 42% of females reported difficulty managing their weight while using contraception, these perceptions often reflected lifestyle factors rather than physiological effects of the contraceptive itself. Clinicians, coaches, and recreationally active females should consider the most appropriate contraceptive type for their circumstances and also reflect on life status and current events as potential contributors to *perceived weight management struggles* or *weight gain*. Individuals may want to consider alternative methods if they are initially affected by DMPA.

## Introduction

In order to prevent pregnancy and manage various medical conditions such as dysmenorrhea, heavy bleeding, endometriosis, and other menstrual disorders many females use hormonal contraceptives. These methods provide daily doses of exogenous hormones, predominantly in the form of oestrogen and progestins, depending on the formulation [1–3]. These hormonal contraceptives contain exogenous steroid hormones that inhibit ovulation and alter endogenous sex hormone concentrations, thereby creating conditions that are unfavourable for pregnancy. Alternatively, other contraceptive methods, such as intrauterine devices (IUDs), either release copper (a non-hormonal option) into the uterus triggering inflammatory reactions toxic to sperm and ova, or release hormones that modify the uterine lining and cervical mucus to prevent pregnancy (Table 1).

The use of modern contraception, particularly birth control, has been documented since the 1960s, with global utilisation continuing to rise across most regions. Recent estimates indicate that in 2022, approximately 874 million women of reproductive age (15–49 years) were using modern contraceptive methods worldwide [4]. In the United Kingdom, approximately 26% of females aged 16–49 years use oral/injectable hormonal contraception [5]. Although evidence is limited in recreationally active females, hormonal contraceptive use appears to increase with physical activity level, as approximately half (49.5%) of elite female athletes report using hormonal contraception [6].

Despite this, many females stop taking oral contraceptives because of factors such as psychological effects and sexual dissatisfaction [7]. Perceived weight change, even when not accompanied by measurable gain, has significant implications for contraceptive adherence and overall health outcomes. Studies have shown that concerns about weight gain are among the most frequently cited reasons for discontinuation or avoidance of hormonal contraceptives [7,8]. Discontinuation because of perceived side effects can increase the risk of unintended pregnancy, compromise reproductive autonomy, and contribute to contraceptive method switching, which may further disrupt menstrual regularity and hormonal stability [9]. Moreover, perceptions of reduced control over weight can negatively affect body image, self-esteem, and engagement in health-promoting behaviours such as physical activity and balanced nutrition. Understanding these perceptions is therefore essential for improving

**Table 1. Overview of contraceptive methods available in the United Kingdom.**

| Contraceptive Method | Active Ingredient | How It Works | Typical Use |
|---|---|---|---|
| **Combined Pill (COC)** | Ethinylestradiol & Levonorgestrel (or other progestins) | • Inhibits ovulation (stops eggs being released from the ovaries)<br>• Thickens cervical mucus<br>• Thins the uterine lining (endometrium) to prevent implantation | Taken daily for 21 days, 7-day break (*can vary depending on brand*). |
| **Progesterone-Only Pill (POP)** | Desogestrel, Levonorgestrel, Norethisterone, etc. | • Thickens cervical mucus<br>• Alters the uterine lining<br>• In some cases, inhibits ovulation (depends on the type) | Taken daily without a break |
| **Hormonal IUD (Intrauterine System)** | Levonorgestrel | • Thickens cervical mucus<br>• Thins the uterine lining (endometrium)<br>• May suppress ovulation in some cases | Inserted by a doctor, lasts 3–5 years |
| **Copper IUD (Coil)** | None – (releases copper) | • Interferes with sperm mobility<br>• Causes inflammation in the uterus, preventing implantation | Inserted by a doctor, lasts 5–10 years |
| **Contraceptive Implant** | Etonogestrel | • Inhibits ovulation<br>• Thickens cervical mucus<br>• Thins the uterine lining (endometrium) | Inserted under the skin, lasts 3 years |
| **Contraceptive Injection** | Depo-Provera (Medroxy-progesterone acetate) (DMPA) | • Inhibits ovulation<br>• Thickens cervical mucus<br>• Thins the uterine lining (endometrium) | Injection every 12 weeks |
| **NuvaRing (Vaginal Ring)** | Ethinylestradiol and Etonogestrel | • Inhibits ovulation<br>• Thickens cervical mucus<br>• Thins the uterine lining (endometrium) | Inserted into the vagina for 3 weeks, removed for 1 week |
| **Patch (Evra)** | Ethinylestradiol and Norelgestromin | • Inhibits ovulation<br>• Thickens cervical mucus<br>• Thins the uterine lining (endometrium) | Applied to the skin weekly for 3 weeks, 1-week break |

contraceptive counselling, adherence, and women's overall well-being. This underscores the need for a mixed-methods approach to capture both the prevalence and lived experience of perceived weight change among contraceptive users.

Previous studies have explored the relationship between contraception and weight change. Most investigations have used quantitative methods, focusing on objective measures such as body mass index (BMI), body composition and/or body-weight [10–12]. Data from 42 studies assessing weight change and combined hormonal contraceptive use demonstrated no substantial weight changes [11]. Additionally, evidence also suggests that most females are at no greater risk of weight gain whilst using combined hormonal oral contraceptive methods [12]. However, very little research has considered individual qualitative experiences of weight gain and contraceptive use.

Although both meta-analyses found no significant effect of contraceptive use on weight gain, most of the included studies lacked an appropriate control or placebo group, limiting the strength of their conclusions [13]. Despite little evidence for the impact of contraception use on weight, a quantitative survey demonstrated that 73% of females living in the UK reported that weight gain was related to oral contraception use [14], highlighting the perceived relationship between contraception use and weight in females. Furthermore, the inclusion criteria in these meta-analyses [11,12] overlooked several other contraceptive methods that do not contain combined oestrogen and progestin formulations, such as depot medroxyprogesterone acetate (DMPA), commonly known as the contraceptive injection. This is important given that DMPA is the only contraceptive method that has been consistently associated with weight gain [15–17].

This oversimplification also overlooks underlying mechanisms such as the potential role that oestrogen plays in appetite regulation. Reviews, including cellular based studies, have demonstrated a role for estrogen in food intake and energy balance mediated by interactions with orexigenic and anorexigenic hormones in the brain [18–20]. Intervention studies have also concluded that oral contraception use may reduce appetite-related hormones associated with satiation, such as cholecystokinin (CCK). After two-to-three months of oral contraception supplementation, CCK was reduced and thus may

be associated with increased appetite and energy intake in females [21]. Despite this, a mismatch between conclusions drawn by literature, mechanisms and the perceived impact of contraceptive use currently exists in females.

Weight change is a multifaceted phenomenon that is often influenced by a range of external factors embedded within everyday life [13]. In females, these complex interactions can interfere with effective weight management and contribute to concerns about weight gain. Such concerns may lead to unnecessary fear, adversely affecting both mental and physical health, and potentially increasing the risk of unintended pregnancy through the discontinuation or avoidance of contraceptive methods [9]. Addressing this issue requires a deeper understanding of the interplay between external influences and individual perceptions related to contraception and weight change.

The aim of this study was to understand perceived weight gain among females of reproductive age and to identify additional contributing factors associated with contraceptive use. By adopting a mixed-methods approach, the study combined quantitative survey data, capturing demographics, contraceptive use, and weight-related perceptions, with qualitative insights to explore individual experiences and perceptions of weight change in the context of contraceptive use.

## Methods

### Study design and philosophical position

This study employed a mixed-methods design, guided by pragmatic philosophy [22]. A pragmatist approach was adopted based on the researchers' views to use the appropriate methods that were going to answer the research question/s. Additionally, in this study, "weight management struggles" were defined as participants' self-reported experiences of attempting to lose weight but perceiving no change on the scales or greater difficulty managing their weight while using contraceptives. This construct was subjective and designed to reflect perceptions, in line with the study's aim to explore female experiences, rather than objectively measured weight change. The term *"weight management struggles"* is used specifically to describe perceived difficulty in controlling or reducing body weight while using contraceptives, whereas *"weight status"* refers to general body-weight outcomes, and *"weight gain"* is reserved for instances where participants explicitly reported an increase in weight.

The mixed-methods design involved collecting quantitative data, which then informed the recruitment and collection of qualitative data interviews. A questionnaire was administered using Jisc Online Surveys. Eligible participants were then contacted to complete a follow-up interview which allowed for an in-depth analysis regarding perceptions of contraceptive use and weight status across the lifespan. Data collection occurred between March 2023 and August 2023. The study and protocol were approved by Liverpool John Moores University Ethics Committee (23/SLN/001) and participants provided written informed consent prior to data collection. Consent explicitly included permission for anonymised quotations to be used in publications and presentations, and all identifying information was removed from transcripts to ensure participant confidentiality. Direct quotes were sent to participants for approval prior to the development of the manuscript.

### Participants

A total of 315 females completed the questionnaire between the dates of 1st June 2023 and 12th August 2023. Participant characteristics are presented in Table 2. Participants were required to be 18 years or older and identify as female from birth. The questionnaire was both voluntary and anonymous. Participants did not receive compensation for participation. The questionnaire was distributed via social media, email, and word of mouth. The advertisement poster contained a QR code, allowing participants to access the questionnaire on their smartphones. Investigators also asked their contacts to forward the questionnaire link and post it in community forums such as online Facebook groups.

Follow-up interviews were conducted between 15th June 2023 and 21st July 2023 with a subset of questionnaire respondents who met eligibility criteria and volunteered to take part. In total, 25 respondents were interviewed; 11 reported it was harder to manage weight, one reported it to be easier to manage weight, and 13 reported that contraceptives did not affect their ability to manage weight (see Table 2).

**Table 2. Participant characteristics for questionnaire respondents.**

| Characteristics | n |
|---|---|
| Age (years) | 31±7 |
| Height (cm) | 166±9 |
| BMI (kg/m²) | 24.5±7.3 |
| Weight (kg) | 67.2±11.1 |
| **Country of residence** | |
| United Kingdom | 298 (96.4) |
| Other | 17 (5.4%) |
| Ireland (n=5), USA (n=3), Spain (n=1) | |
| Australia, Belgium, Canada, Hong Kong, Italy, Kenya, Zambia (n=1 each) | |
| Menstrual status; pre-menopausal/ menopausal | 269 (85.4%)/ 46 (14.6%) |
| **Hours of exercise per week** | |
| Less than 1 | 4 (1.3%) |
| 1-2 | 36 (11.4) |
| 3-4 | 67 (21.3%) |
| 5-6 | 67 (21.3%) |
| More than 6 | 141 (44.8%) |

## Procedure

Jisc Online Surveys (formerly Bristol Online Surveys; Joint Information Systems Committee, Bristol, UK) was used to design and administer the questionnaire, which collected responses anonymously. The questionnaire underwent a two-stage validation process to ensure both face and content validity. Initially, members of the research team (academics in exercise physiology, women's health, and behavioural nutrition) reviewed all items to confirm alignment with the study aims and assess clarity, coherence, and coverage of relevant domains (content validity). The instrument was then pilot tested with five eligible participants to evaluate comprehension, flow, and usability (face validity). Minor wording changes were made based on this feedback to enhance clarity and participant understanding.

Investigators aimed to keep the questionnaire brief to maximise completion rates [23]. The questions were a combination of multiple-choice and free-response items. The online questionnaire took 7±4 minutes to complete and comprised of four sections: (i) General information such as height, weight, demographic, and physical activity levels; (ii) Contraceptive history, (iii) Weight loss, and, the final section asked participants whether they would be happy to be interviewed in relation to the research. Perceived weight management struggles were assessed through a direct self-report item within the questionnaire asking participants whether managing their weight felt *easier, harder, or no different* while using contraception compared with periods when they were not using it. This was designed to capture participants' subjective perceptions of difficulty managing weight in line with the study's focus on lived experience.

During the subsequent timeline interviews, participants expanded on these perceptions, providing context around lifestyle, emotional, and behavioural factors contributing to their experiences. The complete questionnaire instrument can be seen in the Supplementary Materials (S1 File).

Eligible participants were invited to complete semi-structured timeline interviews [24] following contact from a researcher. Interviews were conducted and recorded using online video conferencing software (Microsoft Teams, Redmond, USA). Interviews were conducted in private rooms, with headphones to ensure privacy, providing an accessible and safe space for participants to share their experiences and perceptions. This study sought to explore the wider social, political and environmental contexts of the participants' experiences linked to their use of contraception. The Social Ecological Model [25] underpinned the interview design and analysis, which aimed to understand the multiple layers of personal experience (Fig 1).

Includes the broader social norms, cultural values, and community-level factors that shape the environments in which individuals and organisations operate.

Formal and informal rules, policies, and structures within institutions (e.g. universities, workplaces) that can support or constrain individual behaviours.

Relationships and social networks that can influence behaviour, including those involving family, friends, peers and romantic partners.

Personal characteristics such as biological factors, knowledge, attitudes, beliefs, and behaviours

**Fig 1. Socio-ecological model conceptual framework (SEM) [25].**

Socio-ecological systems theory allows for the consideration of the interrelatedness between systems and how they intertwine to influence and shape females' experiences of weight perceptions whilst taking contraception. The interview questions used can be found in the Supplementary Material (see S2 File).

The timeline-interview method was used to help participants recall and reflect on key events that may have influenced their decision to start using contraception, as well as periods when they experienced weight gain. Participants were first asked to recall when they initially began considering contraception use. From that point, they were guided to discuss any significant life events in chronological order, helping to contextualise their experiences and decisions related to contraception use and weight change.

## Data analysis

Descriptive statistics were generated from the survey data to show the prevalence of contraceptive use and reports of weight gain. To assess whether contraceptive use influenced attempted weight loss, a Pearson's chi-square goodness-of-fit test was used. Statistical significance was set at $p < 0.05$.

Interviews lasted between 29 and 64 minutes (mean = 51 minutes). All interviews were transcribed automatically using Microsoft Teams and later manually checked and refined to ensure scripts were verbatim.

An abductive thematic analysis was conducted using the Social Ecological Model (SEM) to guide coding and theme development. Individual-level influences captured personal beliefs, physiological experiences, and self-directed behaviours, while interpersonal, community, and societal levels reflected broader relational and contextual influences. An abductive thematic analysis [26] was conducted using the Social Ecological Model (SEM) as a framework to guide coding and theme development from interview data. Individual-level influences captured personal beliefs, physiological experiences, and self-directed behaviours. Interpersonal factors referred to relationships and immediate social contexts (e.g.,

partners, peers, coaches) shaping contraceptive experiences and weight perceptions. Community and societal levels encompassed institutional or cultural influences such as workplace demands, university environments, and sociocultural body ideals. This framework allowed multi-layered interpretation of how biological, behavioural, and contextual factors interact to influence perceived weight change. An established six-step reflexive thematic analysis approach [27] was followed. The analyst familiarised themselves with the data and generated initial codes. An abductive process [26] was conducted using the SEM as a framework to analyse the interview data. The overarching levels of the model were used to identify the levels of influence or impact in relation to contraception use, while sub-themes were generated inductively to identify what was occurring within each respective SEM level. Once this initial coding was complete, codes were revisited as part of a focused process to identify potential themes across the data to ensure alignment with research aims. Themes were refined over several iterations by the lead researcher, and a co-author acted as a critical friend to ensure that the themes were defendable and plausible. Finally, the themes were discussed within the research team to ensure credibility and trustworthiness [28].

## Findings and discussion

This section presents both descriptive quantitative survey findings and qualitative interview findings organised through the Social Ecological Model (SEM) to explore females' perceptions of hormonal contraceptive use on weight status and weight management (Table 3). Findings were interpreted through the SEM to highlight how influences on perceived weight management operate across multiple, interacting levels. At the *individual level*, participants described physiological sensations, appetite changes, and emotional responses related to contraception. The *interpersonal level* encompassed support networks and social comparison, such as conversations with partners, peers, or trainers that shaped interpretations of weight gain. The *community level* included transitions into new environments such as university, employment, or sporting settings that altered lifestyle routines. Finally, *societal influences, including* cultural ideals of thinness and public discourse around contraception, framed participants' expectations and self-evaluation. Applying the SEM in this way illustrates that perceptions of contraceptive-related weight change are co-constructed through personal, relational, and structural contexts rather than arising from physiology alone. The results are presented in a way that aligns with the study's analytical

Table 3.  Survey responses and corresponding interview themes on contraception use and weight management.

| Survey Responses (*n* = 104) | Generated Qualitative Themes | Combined Interpretation |
| --- | --- | --- |
| **Harder to manage weight on contraception**: 44/104 (42%) | • **DMPA/injection** repeatedly linked to rapid, hard-to-control gain (Individual)<br>• **Plateau despite training** frustration, reduced motivation (individual/interpersonal). | Interviews corroborated survey patterns and identified DMPA as the primary driver of perceived weight-management difficulty. Supports recommending early monitoring after DMPA initiation and considering alternative methods if >5% gain occurs early. |
| **No difference**: 58/104 (56%) | • **Life transitions** (for example, leaving home or university) linked with poorer diet, increased alcohol consumption, changes in sleep patterns (Interpersonal/Community).<br>• **Irregular eating or disordered eating history** shaping weight across time (Societal/Interpersonal).<br>• **Work or academic stress** reducing activity and altering intake (Community/Societal). | For most participants, lifestyle and psychosocial factors outweighed contraceptive method effects, explaining the survey's "no difference outcome." When viewed through a mixed-methods lens, the significant quantitative result supports the qualitative findings, indicating that perceived weight-management struggles during contraceptive use often reflected lifestyle or contextual influences rather than physiological effects. |
| **Easier**: 2/104 (2%) | • **Stress-related weight loss** or illness, not method-driven (Community/Individual). | Rare reports of "easier" weight loss are not attributable to contraception per se; they reflect non-method stressors. |

Percentages are based on participants who reported attempting weight loss (n = 104). DMPA = depot medroxyprogesterone acetate. Themes are aligned with Social Ecological Model (SEM) levels where applicable.

framework; however, readers may engage in their own naturalistic generalisation [29] and their own unique experience will inform their interpretations of both the data and presented analysis.

The prevalence of contraception over a lifetime was 93%, with 38% of respondents using some form of contraceptive at the time of filling out the questionnaire. Of the 104 participants that had attempted to manage their weight 44 (42%) reported it was harder while using contraceptives compared with periods when they were not. 58 (56%) participants reported no difference, and two (2%) respondents stated it was easier to manage weight (see Fig 2). A chi-square goodness-of-fit test indicated that the distribution of perceived difficulty managing weight while using oral hormonal contraception differed significantly from expected values, $\chi^2(2, N = 196) = 92.86$, $p < 0.001$.

By integrating survey results with interview themes (Table 4), the analysis helped refine and contextualise the quantitative patterns. While 42% of participants initially reported greater difficulty managing their weight while using contraceptives, many later attributed these struggles to lifestyle transitions such as moving away from home, irregular eating patterns, or stress when reflecting during timeline interviews. This highlights the value of combining methods, as survey responses alone may overestimate the direct impact of contraceptives on weight. Qualitative insights clarified that lifestyle and psychosocial factors often explained these perceptions, while also identifying specific cases, such as those involving DMPA, that warrant closer examination.

## Contraception use

Out of the 25 participants who were interviewed, 11 reported experiencing difficulty with weight status and weight management whilst using contraception, 13 reported no differences and one participant found it easier to lose weight whilst using

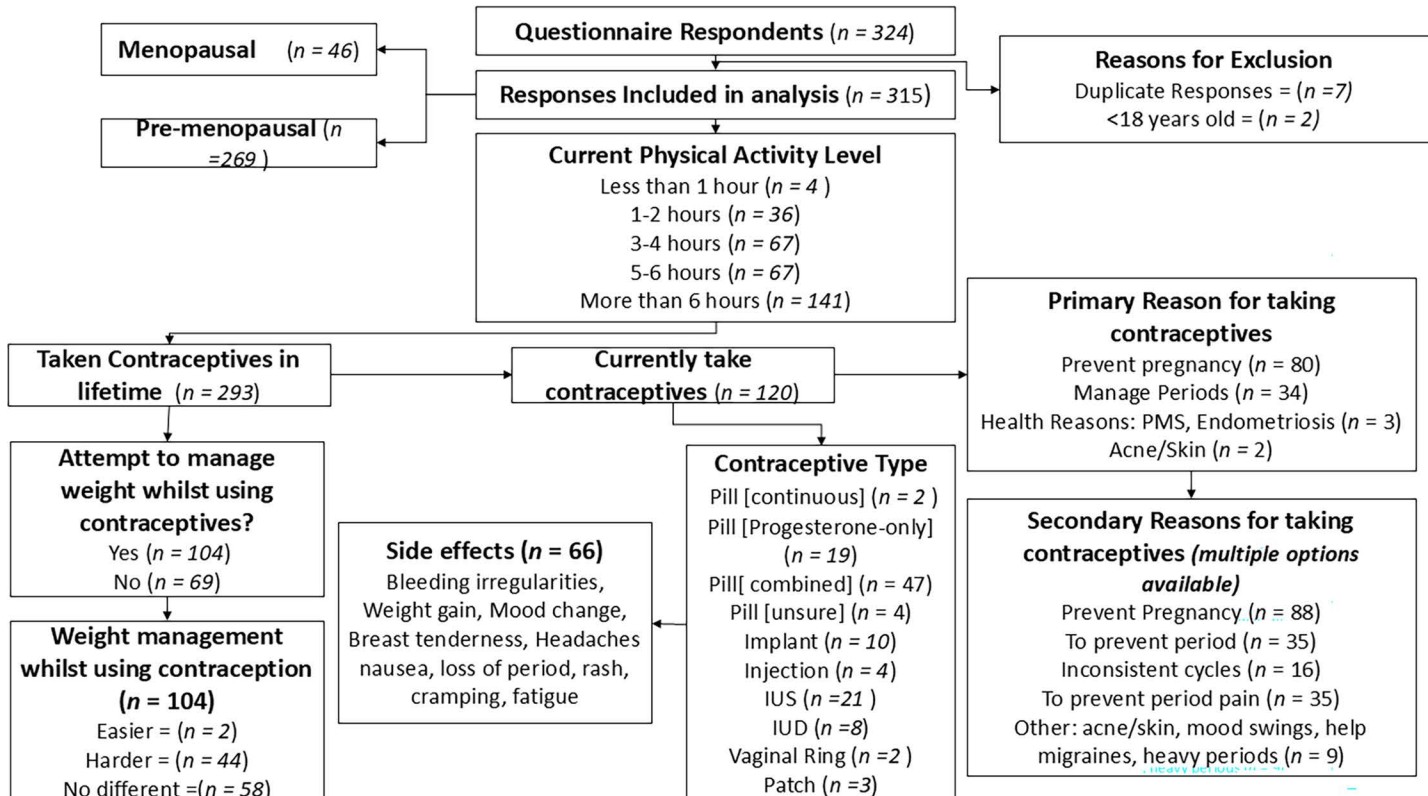

**Fig 2. Prevalence, contraceptive history, contraceptive type, side effects and weight management capabilities responses from questionnaire.**

**Table 4. Participant characteristics for interviewed individuals.**

| Characteristics | n |
|---|---|
| Age (years) | 31.8±5.9 |
| Height (cm) | 165.7±8.0 |
| BMI (kg/m²) | 23.2±6.1 |
| Weight (kg) | 63.3±17 |
| **Country of residence:** | |
| United Kingdom | 22 (88.0%) |
| Other USA (*n*=1) Canada (*n*=1) Ireland (*n*=1) | 3 (12.0%) |
| **Hours of exercise per week** | |
| 3-4 | 5 (12.0%) |
| 5-6 | 5 (24.0%) |
| More than 6 | 14 (64.0%) |
| Harder to lose weight on HC | 11 (28.0%) |
| Easier to lose weight on HC | 1 (12.0%) |
| No difference in weight loss | 13 (60.0%) |

contraception. However, following the interviews, eight out of the original 11 participants who initially reported difficulties later attributed these challenges to other factors (see below). The DMPA injection was consistently reported to contribute to *weight management struggles* and difficulty losing weight. In contrast, inconsistencies were reported in weight status in other contraception types where individuals were either unaffected or experienced difficulties. Participants frequently described the DMPA injection as a key factor contributing to difficulties with weight management, stating the following:

> *"But I was going to the gym. I'm not joking like I was I couldn't get my head round for probably about six months and it did start to damage my mental health, actually because it was kind of like a couldn't get my head round what and why I couldn't lose the weight and it we're not talking like, I mean, so at the moment, I'm yeah, probably about 68 kilos, my lightest I was 58. I'd quite like to lose 5, but I it's really difficult and I so basically, I was in the gym one day and I've got to know I was in the gym all the time. I was doing weights. I was doing cut lot of doing a real combination consistently.*

> *One of the PTs said to me here is that I don't wanna overstep the mark, but he was like, have you thought about your hormones. And I was like, never really kind of thought about it like that.*

> *And he was like, yeah, I trained a few women who've had that problem, went home, Googled it, turns out, apparently the depo injection is horrendous and all these other found all these forums that were like, you know, people that were PTs, way fitter than I was and, like, couldn't lose the weight." P17*

One participant described her experience of using the DMPA injection in way that she couldn't control, she recounts

> *"The injection, the injection, I felt like I was just ballooning"* P14

Whilst the depot medroxyprogesterone acetate (DMPA) is clinically appealing with its extremely low annual failure rate (<1%), previous research has identified that ~20% of users experience >5% body-weight increase within the first 6 months of DMPA. In addition, those who experience >5% weight gain were predicted to have continuous excessive weight gain whilst being administered DMPA [30,31]. This in turn could explain why many of the participants in the current study who had DMPA reported issues with their weight and did not attribute it to other potential lifestyle factors. This is an important

consideration for clinicians, who should monitor weight changes during the first six months of DMPA administration. Users who experience early weight gain should be counselled about alternative contraceptive options to minimise the potential long-term health risks associated with overweight and obesity.

It is important to note that, for some females, perceived weight gain corresponds with actual weight change, particularly among users of the implant or DMPA [16]; however, this observation, together with the findings of the present study, should not be interpreted as evidence of a universal or causal effect. Although metabolic markers were not assessed in this study, previous research has identified plausible biological mechanisms by which DMPA may influence weight status. DMPA has been associated with increases in circulating leptin levels, which may reflect leptin resistance, thereby impairing satiety signalling and promoting increased energy intake [32]. In addition, progestins such as medroxyprogesterone acetate can alter hypothalamic neuropeptides, including neuropeptide Y (NPY) and agouti-related peptide (AgRP), both of which stimulate appetite, while lacking the oestrogenic component of combined contraceptives that can exert anorexigenic effects through hormones such as cholecystokinin (CCK) [18–20]. These alterations in appetite-regulating pathways provide a biologically plausible explanation for the consistent reports of weight gain and difficulty losing weight among DMPA users in this study.

It is important to interpret these DMPA-related findings within the context of the study's methodological and contextual limitations. The current data are based on retrospective self-report and therefore may be influenced by recall bias, particularly regarding the timing and extent of weight changes. Perceived weight gain may not always align precisely with objective measurements, and prior prospective studies provide mixed evidence. For example, research reported that over a 36-month period, DMPA users experienced significantly greater increases in body weight (+5.1 kg), fat mass (+4.1 kg), and central-to-peripheral fat ratio compared with oral and non-hormonal contraceptive users, suggesting a potential biological influence of DMPA on adiposity [15]. In contrast, Bonny et al., [31] examined adolescents initiating DMPA and found that early weight gain (>5% within six months) predicted continued excessive weight gain at 12–18 months. While this provides valuable insight into early predictors of weight gain, it should be noted that this study was conducted in adolescents, whose hormonal and behavioural profiles differ from adult women in the present study. Such discrepancies between studies likely reflect differences in participant age, duration of use, and individual metabolic or lifestyle factors. Overall, the current findings should therefore be viewed as reflecting a multifactorial process in which both pharmacological effects and coexisting lifestyle behaviours contribute to perceived weight change.

Whilst the results of this study highlight the potential effects of contraception, particularly DMPA, on weight management struggles and perceived difficulty in losing weight, these findings are based on self-reported experiences and may be subject to recall bias, particularly as some participants were asked to reflect on contraceptive use many years earlier. The association between DMPA and weight change reported in this study reflects participants' subjective interpretations rather than causal physiological evidence. Moreover, DMPA remains a highly effective (>99%) and widely used contraceptive with a well-established clinical safety profile, and nothing in the current findings challenges its overall efficacy or suitability for many users. It was also evident that many participants identified additional contributing factors that affected their weight status alongside contraceptive use during the timeline interviews. These themes included moving out of the family home, disordered eating patterns or eating disorders, and life stressors such as work or family pressures. These factors are explained in detail in the sections below.

### Moving away from home

Out of the 25 participants who took part in this study, 15 reported attending University or college and thus, moving away from the family home. This represents the interpersonal level of the SEM, as it highlights the shift from a structured home environment to an independent setting where external influences, such as peer behaviors, social norms, and new lifestyle habits, begin to shape dietary and health-related decisions. During this period, participants described major changes in their diet and lifestyle, transitioning from healthy home-cooked meals and regular sleep to late nights, frequent takeaway food,

and higher alcohol consumption. Research suggests that lifestyle changes occur in the first year of university or college [33], often resulting in a mix of weight gain, maintenance and weight loss as a result of these changes [34]. Of the 15 participants who attended university, eight acknowledged having a poor diet, and three indicated alcohol consumption was high.

*"That was like 1 meal a day. A lot of alcohol. I smoked a lot so that I know that don't count as a diet, but that was my diet. Yeah. Really, really poor. I woke up in time for lecture and was running out the door or wherever, and I was really young. I wouldn't get a bed or. Yeah, the typical first year uni kind of student was me And you know, it wasn't intentional, Like I wasn't avoiding food for to look at certain way or keep the weight off anything. I just was forgetting or which is really weird for me, was forgetting or just didn't have the time and then I'd make sure I had the time when I got back in the evening."* P14

*"And then also going out like Wednesday night, Friday night, Saturday night and going to bed at four o'clock. And honestly, I don't know how I did it. Now I have one late night and I'm like, I need a month off". I definitely remember having a giant box of ice cream in the freezer and would get it out and eat it with a spoon the top. And eating cheese and crackers like it was going out of fashion"* P18

*"And at Uni I was doing the usual uni things going out, partying, saving most of my money to spend on that kind of stuff rather than eating"* P19

*"For the very first time ever it was space of about six weeks, and then I went to university in, I think the freedom, the having my own money and being able to decide what I eat when I eat. It made me put that weight on back very quickly and actually gained more than in the end."* P8

The current study demonstrates that those who went to university or college experienced lifestyle adjustments resulting in a higher alcohol consumption, unhealthy dietary changes, and disrupted sleep patterns, which are consistent with previous research [35,36]. It is therefore reasonable to suggest that lifestyle factors, such as moving away from the family home and transitioning to independent living, have significant effects on overall health behaviour and weight status. Previous research also indicates that adolescent and young adult females often report a desire to be thinner and experience anxiety related to body shape, which may further influence eating habits during this period of life.

### Relationships with food (disordered eating)

When the 25 participants were asked about their typical nutrition while on contraception, 20 mentioned that their diets were "poor" in terms of quantity or quality and "could have been better." Of these participants, four mentioned experiencing disordered eating or an unhealthy relationship with food at some point in their lives. This aligns with the SEM, particularly within the societal level, where broader cultural expectations, peer influence, and media portrayals of body image contribute to unhealthy relationships with food. Research indicates that adolescents and young females commonly report a desire to be thinner and experience anxiety regarding body shape [37–39]. Poor relationships with food that develop during early adolescence and childhood often persist into adulthood. These early influences, shaped by societal beauty standards, peer comparisons, and family attitudes toward weight and diet, reflect how interpersonal and sociocultural factors contribute to disordered eating behaviours. Additionally, longitudinal research showed that the presence of eating disorder symptoms during adolescence significantly increases the risk of similar symptoms in adulthood [40].

The following quotes come from participants describing their diet, with behaviour consistent with disordered eating in order to maintain a certain body size.

*"It was quite restrictive and I remember people pointing out how thin I was, once I'd hit that point somehow it kind of spiralled into binge eating. So I was being restrictive, but then eating a lot and in secret."* P28

*"I probably underate. I definitely was that sort of classic teenager worried that I needed to lose weight and I definitely didn't, and yeah, I used to often sort of skip lunch. And not have much of a breakfast, and then often wouldn't have lunch."* P18

The participants that mentioned a sub-optimal diet described it as "a typical university diet," characterised by high alcohol consumption consistent with previous research [41,42] and convenience foods. A common theme mentioned was a lack of knowledge or experience cooking and food preparation, which often led participants to either under-eat, due to not knowing how to prepare food properly, or over-eat due to the accessibility and convenience of typically lower-quality foods like frozen or fast food. This highlights the importance of establishing healthy dietary habits in life, as this can significantly impact long-term health outcomes [43].

Transitioning to university marks a crucial period where individuals take on greater responsibility for their food choices and healthy lifestyle practices [44] however, many young adults may lack the experience in food shopping, meal preparation, and planning [45].

*"My diet when I was a kid was pretty poor. It was really bad. A lot of junk food and it was a lot like Iceland, like cheap frozen food and stuff like that, so it's quite poor."* P13

The following quote depicts a participant's perception of their own weight fluctuation.

*"I had an eating disorder when I was younger. And then the first year of university was a bit all over the place because I couldn't really work out, basically my parents until I left home had been in control of what I ate. So when I go to university, I was like I don't know how to do this. And so I think my weight went up and then went down because I was really restrictive."* P16

Of the 20 participants who reported having a poor diet, seven stated that it was harder to lose weight while on contraception, twelve believed it made no difference, and one found it easier. The vast majority of participants interviewed described experiencing irregular eating patterns at some point in their lives. These irregular eating habits, which often coincided with contraceptive use, may have contributed to changes in weight status rather than being directly caused by the contraceptives themselves.

Participants typically started taking contraceptives around the same time as major life transitions, such as moving away from home for the first time or entering relationships. These circumstances can significantly alter a person's habitual diet and therefore weight status. Because these events often occurred simultaneously with the initiation of contraception, weight changes were easily attributed to contraceptive use, even though lifestyle factors were likely to play a substantial role.

## Life stressors

Four of the eleven participants who initially reported difficulty losing weight while on contraception identified environmental and life stressors as the primary contributors to their weight gain. These factors included stressful academic or work environments, reduced physical activity due to work demands, and other life events. This aligns with both the *societal* and *community* levels of the SEM, as work-related pressures, stressful environments, and institutional expectations can influence individuals' capacity to manage their health and weight. Evidence of this can be found in the literature [46,47].

One participant attributed her weight gain not to the contraceptive implant but to the stress of completing her PhD thesis and poor eating habits:

*"And I've put on probably half a stone, maybe a stone, since I've had it fitted (contraceptive implant), but that is I'm gonna say purely because of the stress of getting my PhD thesis in like the long hours, and I thought if I'm miserable, I might as well be miserable eating something.*

*Back at the time, I would have said Ohh yeah, having the injection or being on the pill that's made me fat that's made me gain weight and then the little voice in my head of the nurse kind of rings in my ears and she's like, well, if you're eating more then yeah, obviously gonna put on weight.*"P14

Other participants reported that work-related factors played a significant role in their weight gain, often overshadowing the influence of contraception. The nature of their jobs, including changes in activity levels and nutritional choices, contributed to their weight changes.

One participant described how her work at McDonald's led to poor dietary habits and subsequent weight gain.

*''I got my very first job at McDonald's between in the summer in between going to Uni, and finishing sixth form. So for me it was like a like a theme park. Let's try everything and you know, and then the people who worked there for a while. They were like, why don't you try adding this on top of this burger or having this? So my nutritional choices were quite appalling, to be honest.... there were days when I had McDonald's three times a day and quite often.*" P8

Another participant (P28) faced similar challenges, attributing her weight gain to decreased activity levels due to a desk-based job rather than her contraceptive use:

*"And I have wondered if the coil has made me gain weight, but then I don't know if that's. I think because I've been so inactive compared to my usual activity levels, it's probably that."*

*"I sort of hadn't really linked the contraception with weight gain at that point. But I'd linked it more with the fact I was doing a desk-based job, so my activity decreased, I gained some weight, but I linked it with that so I don't know if that's right or wrong."*

Five of the thirteen participants who reported no noticeable changes in their ability to lose weight while using contraception also identified life stressors as influencing their body weight. For instance, one participant who consistently used the combined pill described losing weight due to stress and gaining weight during periods of travel, when her routines became inconsistent.

*"I had my first serious breakup, and I lost a lot of weight from that. Just like stress and stuff, and I had a few years of turmoil during that sixth form."* P9

*"I was pretty much away for nine months, and during that period, I was completely random with the pill and I put on a lot of weight, yeah... Definitely in Greece, quite a bit of drinking because you got all your alcohol free, and you'd go for these big group meals where you ate for free."* P9

Stress from significant life events often led to weight gain for some participants. The following quote captures this experience:

*"And I put on a lot of weight in second year. I stopped training really at all... My housemate went missing for five weeks and then was found dead. And then my boyfriend at the time, his mum got diagnosed with cancer. Then his best mate got hit by a van and killed. And then my sister got epilepsy and was having seizures every other day out of nowhere. And that all kind of happened within a few months of each other. And it was just stressful."* P18

These narratives highlight the complex interplay between contraception use and life stressors such as work, stress, lifestyle changes, and major life events in influencing *perceived weight management experiences*. While some participants

initially attributed weight changes to contraception, a closer examination revealed that life stressors often played a more substantial role. Additionally, there is little evidence supporting a direct link between oral contraceptive use and weight gain when DMPA injection is not factored. This should be a major consideration when reviewing the research [9,11,48]. It is therefore important to adopt a holistic approach that considers an individual's overall lifestyle, health behaviours, and external circumstances when evaluating the potential impact of contraception on weight. A summary of these findings, integrated with the Social Ecological Model, is presented in Fig 3.

## Limitations

Whilst the present study provides valuable insights into the perceived effects of contraception on weight status, it is important to acknowledge its limitations. The cross-sectional design of the study limits the ability to establish causality between contraceptive use and perceived weight changes. Although existing evidence suggests that perceived changes in weight often reflect actual changes in weight [16], a longitudinal design following participants over time could provide more definitive evidence regarding the temporal relationship between contraceptive use and weight management. In addition, whilst the timeline interviews enabled an in-depth exploration of an individual's experiences alongside their contraceptive history, the findings heavily relied on retrospective self-reported data. Participants were asked to recall events (such as the first time they took contraceptives); for some this spanned 20 years. The accuracy of this recollection may be affected by recall bias, memory degradation, and the influence of more recent experiences, potentially reducing the reliability of some responses.

A further limitation is the representativeness of the sample. Participants were predominantly based in the UK and highly active, with 44.8% reporting >6 h of exercise per week. This profile likely reflects a health-conscious population who may be more aware of or sensitive to weight changes than the general population. As such, findings should not be generalised to all females of reproductive age but may be most applicable to physically active or health-orientated groups. Future research should aim to recruit more diverse and representative samples to examine whether perceptions differ among less active populations and across different cultural contexts.

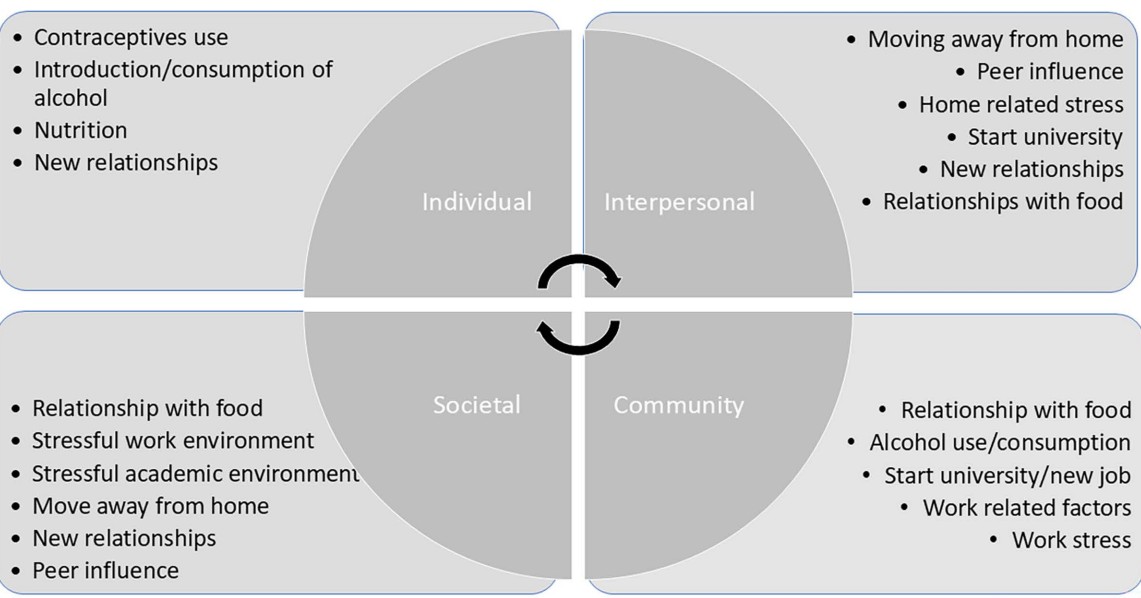

**Fig 3. Factors affecting weight status and weight management in females presented in socio-ecological levels.**

Finally, the operationalisation of "weight management struggles" was based on subjective perceptions (for example, attempts to lose weight without perceived change on the scales) rather than objectively measured weight change. These findings should therefore be interpreted as reflective of perceptions rather than physiological outcomes.

## Conclusion and recommendations

The aim of this study was to evaluate perceived weight gain and weight-management difficulties among females of reproductive age taking or have taken contraceptives. The secondary aim was to identify any additional contributing factors, beyond contraception use, that may have influenced these perceptions. This is an important area, as research is limited, and misconceptions about the effects of contraceptives on weight can influence females' contraceptive choices and overall well-being, leading to unnecessary anxiety or discontinuation of effective birth control methods, putting them at risk of unplanned pregnancy. Our findings do not establish causal effects of any contraceptive method on weight status. DMPA remains a clinically effective and safe contraceptive option, and perceived changes should be interpreted within the broader behavioural and contextual factors highlighted in this study.

The Socio-Ecological Model (SEM) proved to be a valuable framework in understanding the complex interplay between individual, interpersonal, and environmental factors influencing weight management perceptions among females using contraceptives. It helped highlight the multiple layers at which these perceptions are shaped, from personal behaviours to broader societal influences. Among the 315 respondents, 42% of females reported that they struggled with weight management whilst taking contraceptives; however, this should be interpreted with caution as participants were predominantly UK-based and highly active. When timeline interviews were conducted and participants reflected on broader life events, many concluded that external lifestyle factors such as moving away from home, relationships with food, and life stressors may have been more influential than contraception itself. Because this study relied on retrospective self-report, many participants could not recall the specific name or formulation of their contraceptive, which limited precise identification of progestin types However, it is important to note those who received depot medroxyprogesterone acetate (DMPA) which has a uniform composition perceived the contraceptive as contributing to *weight management struggles* and, in some cases, weight gain, independent of other confounding variables. DMPA may influence weight status through its effects on appetite-regulating hormones and energy balance. DMPA use has been associated with elevated leptin levels, a hormone involved in satiety signalling; however, paradoxically, this may reflect leptin resistance, where the brain no longer effectively responds to leptin's satiety cues, potentially increasing food intake [32]. Furthermore, DMPA may alter neuropeptides within the hypothalamus, including neuropeptide Y (NPY) and agouti-related peptide (AgRP), which are known to stimulate appetite. Preclinical models have demonstrated progestins' ability to upregulate these orexigenic pathways, suggesting a biological mechanism through which DMPA may contribute to increased appetite and subsequent weight gain [18,19]. Unlike combined oral contraceptives, which contain oestrogen that can suppress food intake through modulation of anorexigenic signals such as cholecystokinin (CCK), DMPA lacks oestrogen and may therefore be devoid of these appetite-suppressing effects [20,21]. This distinction is particularly relevant for athletes or individuals with weight-sensitive performance goals, emphasising the need to monitor early weight changes and consider alternative contraceptive options when necessary.

While the present study explored lifestyle influences qualitatively, objective assessment of diet quality and activity levels was not feasible due to the recall bias inherent in retrospective assessment. Importantly, women's perceptions of how contraception affects weight are central to real-world adherence, influencing discontinuation, inconsistent use, and method switching. The present mixed-methods design was not intended to determine mechanistic pathways, and findings should not be interpreted as biomedical confirmation that DMPA leads to weight gain. Understanding these perceptions is therefore as important as, if not more important than, measuring physiological changes. These lived experiences are essential for improving contraceptive counselling and supporting sustained use. Future prospective studies should complement these findings with objective behavioural and physiological measures collected in real time.

Finally, these findings reflect the characteristics of the current sample, which was predominantly UK-based and composed largely of physically active females. As such, the results may be most relevant to health-conscious or active populations and should not be assumed to generalise to all females of reproductive age. Future research should further investigate both the mechanistic effects of DMPA on appetite regulation and energy balance and explore whether similar perceptions are found in more diverse and representative female populations across cultural contexts.

## Supporting information

**S1 File. Questionnaire completed by respondents.**
(DOCX)

**S2 File. Timeline online interview questions.**
(DOCX)

## Author contributions

**Conceptualization:** Stephen McQuilliam, Elizabeth Mahon, Amy Whitehead, Kelsie Olivia Johnson.

**Data curation:** Elizabeth Mahon, Amy Whitehead, Kelsie Olivia Johnson.

**Formal analysis:** Elizabeth Mahon, Amy Whitehead, Kelsie Olivia Johnson.

**Investigation:** Tamara Prostináková, Kathelijne Gabrielle Silang, Alyssia Griffiths-Gray, Amy Whitehead, Kelsie Olivia Johnson.

**Methodology:** Leonie Bass, Tamara Prostináková, Kathelijne Gabrielle Silang, Alyssia Griffiths-Gray, Amy Whitehead, Kelsie Olivia Johnson.

**Project administration:** Leonie Bass, Tamara Prostináková, Kathelijne Gabrielle Silang, Alyssia Griffiths-Gray, Amy Whitehead, Kelsie Olivia Johnson.

**Resources:** Kelsie Olivia Johnson.

**Software:** Kelsie Olivia Johnson.

**Supervision:** Amy Whitehead, Kelsie Olivia Johnson.

**Validation:** Kelsie Olivia Johnson.

**Writing – original draft:** Leonie Bass, Elizabeth Mahon, Kelsie Olivia Johnson.

**Writing – review & editing:** Leonie Bass, Tamara Prostináková, Kathelijne Gabrielle Silang, Elizabeth Mahon, Amy Whitehead, Kelsie Olivia Johnson.

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
