## [Decision Letter · Decision Letter 0]

12 Aug 2025

PONE-D-25-30751Does it hold weight? The perceived effects of contraceptive use on weight status in females: A Mixed-Methods StudyPLOS ONE?

Dear Dr. Johnson,

Thank you for submitting your manuscript to PLOS ONE. After careful consideration, we feel that it has merit but does not fully meet PLOS ONE’s publication criteria as it currently stands. Therefore, we invite you to submit a revised version of the manuscript that addresses the points raised during the review process.

We look forward to receiving your revised manuscript.

Kind regards,

Vandana Dabla

Academic Editor

PLOS ONE

Journal Requirements:

2. Please ensure that you refer to Figure 1 and 3 in your text as, if accepted, production will need this reference to link the reader to the figure.

3. We note that there is identifying data in the Supporting Information file “Data Submission.xlsx”. Due to the inclusion of these potentially identifying data, we have removed this file from your file inventory. Prior to sharing human research participant data, authors should consult with an ethics committee to ensure data are shared in accordance with participant consent and all applicable local laws.

-Location data

Additional Editor Comments:

**You are requested to read the reviewers comments carefully and submit the point wise reply against each raised query. For comments you do not agree and feel that the original submission is justified reference to your study, must also be replied in detail.  **

Reviewers' comments:

Reviewer's Responses to Questions

**Comments to the Author**

1. Is the manuscript technically sound, and do the data support the conclusions?

Reviewer #1: Partly

Reviewer #2: Partly

2. Has the statistical analysis been performed appropriately and rigorously?

Reviewer #1: N/A

Reviewer #2: N/A

3. Have the authors made all data underlying the findings in their manuscript fully available?

Reviewer #1: Yes

Reviewer #2: No

4. Is the manuscript presented in an intelligible fashion and written in standard English?

Reviewer #1: Yes

Reviewer #2: Yes

Reviewer #1: Major Comments

1. Mixed-Methods Integration Gap:

o While the authors adopt a sequential explanatory design (quantitative followed by qualitative), the integration of findings across methods is largely descriptive and lacks depth. There is limited evidence of how the qualitative interviews build upon or help explain specific patterns observed in the survey data.

o Recommendation: Clearly articulate in the results and discussion sections how the qualitative themes help to confirm, challenge, or explain the quantitative findings. Consider using a joint display or matrix to demonstrate this integration explicitly.

2. Sampling and Representativeness:

o The study claims to generalize to "females of reproductive age," but the sample is predominantly from the UK and appears to over-represent physically active women (e.g., 44.8% report >6 hours of exercise/week). This could bias perceptions of weight management and may not reflect the general population.

o Recommendation: Discuss the sampling bias and its implications more clearly in the limitations section. Consider tempering generalizations and clarifying that findings may be most relevant to physically active, health-conscious populations.

3. Operationalization of “Weight Management Struggles”:

o The manuscript heavily relies on participants’ self-reported struggles with weight loss or gain without defining how "struggle" was measured or whether this referred to actual weight change or subjective difficulty.

o Recommendation: Provide clarity in the methods on how “struggle” was conceptualized and whether validated scales or anchor questions were used. Otherwise, the construct remains vague and open to interpretation.

4. Interpretation of DMPA-Related Weight Gain:

o While DMPA is repeatedly identified as associated with weight gain, the discussion lacks nuance regarding dose, duration, and metabolic factors. The attribution is largely anecdotal and may reflect confirmation bias.

o Recommendation: Strengthen this section by discussing the limitations of self-reported weight change, the potential for recall bias, and contrasting findings from prospective studies on DMPA. Consider the role of coexisting lifestyle changes.

5. Use of Social Ecological Model (SEM):

o Although SEM is introduced as the framework, its application in the findings is inconsistent. The mapping of themes onto SEM levels (e.g., interpersonal, societal) seems forced at times and lacks theoretical depth.

o Recommendation: Ensure that SEM is systematically applied across all themes with clearer justification. Avoid vague attributions (e.g., "societal level" for stress) and explain how each level influenced contraceptive perception and behavior.

Minor Comments

1. Background in Introduction:

o The introduction would benefit from a stronger contextual background on the significance of perceived weight changes in contraceptive users, especially linking this perception to contraceptive adherence and health outcomes. A more focused rationale would help set up the need for this mixed-methods study.

2. Study Location Missing in Abstract:

o The abstract should clearly state where the study was conducted (e.g., United Kingdom). This adds context and helps readers assess the generalizability of findings.

3. Repeated References:

o Some references appear more than once with minimal distinction (e.g., Brown & Clegg 2010a and 2010b; Gallo et al. 2014a and 2014b). Please check for redundancy and consolidate citations where appropriate.

4. Outdated Prevalence References:

o The introduction cites contraceptive prevalence using data from older sources (e.g., WHO 2014, Alkema et al. 2013). Consider updating this with more recent national or global estimates (e.g., WHO or UNFPA 2022 reports).

5. Questionnaire Validation:

o While pilot testing is mentioned, there is no clear description of how the questionnaire was validated. Was face validity or content validity assessed formally? Clarifying this would strengthen the reliability of the quantitative data.

6. Abbreviation Use (e.g., DMPA):

o Ensure that all abbreviations are written out in full at first mention (e.g., “DMPA” should be “Depot Medroxyprogesterone Acetate (DMPA)” initially). This helps readers who are not familiar with contraceptive terminology.

7. Title Clarity:

o The main title “Does it hold weight?” is catchy but somewhat vague. Consider adding a more descriptive subtitle to improve clarity (e.g., “Perceived Effects of Contraceptive Use on Weight in Females: A Mixed-Methods Study”).

8. Grammar and Formatting:

o Minor grammatical issues and typographical errors are scattered throughout the manuscript (e.g., missing articles, extra spaces, inconsistent tenses). A thorough proofread is recommended.

9. Inconsistent Terminology:

o The manuscript uses terms like “weight gain,” “weight status,” and “weight management struggles” interchangeably. These should be clearly defined and used consistently to avoid confusion.

10. Figures and Tables:

• Figure references (e.g., “FIGURE 2 NEAR HERE”) appear as placeholders. Ensure figures are properly inserted, numbered, captioned, and clearly explained in the text. Table formatting could also be improved for clarity.

11. Use of First-Person Language:

• Phrases such as “we are aware” appear in the manuscript. Consider revising these to maintain a neutral academic tone unless first-person language is permitted by journal guidelines.

12 Consider including basic inferential statistical analysis, such as calculating odds ratios or chi-square tests, to enhance the quantitative interpretation of survey findings. If appropriate, consult a statistician to strengthen the analytical rigor.

Reviewer #2: Review Comments to the Author:

Thank you for the opportunity to review this manuscript. The topic - perceived effects of contraceptive use on weight status in females - is both clinically relevant and timely, especially in the context of informed contraceptive choice and body image concerns among women.

The authors have successfully applied a mixed - methods design to gain both breadth and depth in understanding women’s experiences. The integration of qualitative interviews, grounded in a socio - ecological model, provides valuable insight into the broader psychosocial and lifestyle factors that interact with contraceptive use and perceived weight changes.

However, there are several important methodological and clinical issues that should be addressed to improve the clarity, rigor, and reproducibility of the study:

Lifestyle Confounding Factors Not Assessed:

The quantitative survey did not include questions about nutritional patterns (e.g., meal frequency, diet quality) or detailed physical activity classification (e.g., light/moderate/vigorous activity). These omissions are significant given the study’s focus on weight perception and management.

No Objective Measures of Body Composition:

The study relies entirely on self-reported weight-related perceptions without data on body composition (fat mass vs. muscle mass) or clinical indicators (e.g., presence of edema or fluid retention, which are common with some contraceptives). This limits the physiological interpretability of the findings.

Missing Pharmacological Details:

The manuscript does not specify which types of progestins were included in the combined oral contraceptives (COCs) used by participants. Since progestins have differing metabolic effects, this information is critical and should be reported.

Clinical Inconsistency Not Addressed:

One participant reportedly used COCs to stop migraines, which contradicts current clinical guidelines that contraindicate combined hormonal contraceptives in women with migraine-especially with aura - due to elevated stroke risk. This discrepancy should be clarified or contextualized.

Anonymity of Qualitative Data:

While the interview excerpts are rich and informative, some are highly personal and may allow participants to be identified. It should be clarified whether participants gave explicit consent for their quotations to be published.

Recommendations for Revision:

-Include specific details about contraceptive formulations (especially the progestins used).

-Clarify ethical procedures regarding qualitative data publication.

-Add discussion about the lack of objective anthropometric or clinical data (e.g., edema).

-Consider a stronger discussion of lifestyle confounders and suggest their inclusion in future research tools.

-Clarify or revise the mention of COC use for migraine management to align with safety guidelines.

**Do you want your identity to be public for this peer review?** For information about this choice, including consent withdrawal, please see our Privacy Policy

Reviewer #1: No

Reviewer #2: No

---

## [Author Response · Author response to Decision Letter 1]

22 Oct 2025

Reviewers Comments

Your Comment / Recommendation Revision

All of the formatting for this manuscript has now been aligned to PLOS ONE’s style this includes:

• Cover Page

• Referencing (numerical)

• Fig labels in-text

• Fig labels diagram

• Table Formatting

2. Please ensure that you refer to Figure 1 and 3 in your text as, if accepted, production will need this reference to link the reader to the figure. Figures have been added to the text

We note that there is identifying data in the Supporting Information file “Data Submission.xlsx”. Due to the inclusion of these potentially identifying data, we have removed this file from your file inventory. Prior to sharing human research participant data, authors should consult with an ethics committee to ensure data are shared in accordance with participant consent and all applicable local laws.

-Location data

Additional Editor Comments:

You are requested to read the reviewers comments carefully and submit the point wise reply against each raised query. For comments you do not agree and feel that the original submission is justified reference to your study, must also be replied in detail.

All data which contains identifiable information has now been redacted.

1. Mixed-Methods Integration Gap:

o While the authors adopt a sequential explanatory design (quantitative followed by qualitative), the integration of findings across methods is largely descriptive and lacks depth. There is limited evidence of how the qualitative interviews build upon or help explain specific patterns observed in the survey data.

o Recommendation: Clearly articulate in the results and discussion sections how the qualitative themes help to confirm, challenge, or explain the quantitative findings. Consider using a joint display or matrix to demonstrate this integration explicitly. We thank the reviewer for this constructive observation. In line with your recommendation, we have strengthened the integration of the quantitative and qualitative strands in both the Results and Discussion. Specifically:

1. Results section – We now present survey findings alongside corresponding interview insights, and have added a joint display table (Table 3) that aligns survey responses with emergent qualitative themes and provides an integrated interpretation. This makes explicit how interview data confirm, expand upon, or challenge survey patterns (e.g., survey reports of weight management difficulties clarified as lifestyle-related in interviews, with DMPA consistently identified as an exception).

2. Discussion section – We revised the opening paragraphs to interpret these integrated findings, highlighting how mixed-methods analysis refined the survey results. While 42% initially reported greater difficulty with weight management on contraceptives, qualitative reflection suggested many of these difficulties were attributable to lifestyle and psychosocial factors. However, interviews consistently highlighted DMPA as a direct driver of perceived weight struggles. This integration demonstrates how qualitative data added depth, nuance, and explanatory value beyond descriptive survey results.

We believe these revisions substantially enhance the coherence and depth of our mixed-methods design and directly address the reviewer’s request for clearer articulation of how qualitative themes build upon and explain quantitative findings.

2. Sampling and Representativeness:

o The study claims to generalize to "females of reproductive age," but the sample is predominantly from the UK and appears to over-represent physically active women (e.g., 44.8% report >6 hours of exercise/week). This could bias perceptions of weight management and may not reflect the general population.

o Recommendation: Discuss the sampling bias and its implications more clearly in the limitations section. Consider tempering generalizations and clarifying that findings may be most relevant to physically active, health-conscious populations. We agree with the reviewer’s observation and have clarified the representativeness of our sample.

Limitations section - We now explicitly acknowledge that participants were predominantly UK-based and highly active, with nearly half reporting more than six hours of exercise per week. We note that this likely reflects a health-conscious population and that findings should not be assumed to generalise to all females of reproductive age. The following paragraph has been added: “A further limitation is the representativeness of our sample. Participants were predominantly based in the UK and highly active, with 44.8% reporting more than 6 hours of exercise per week. This profile likely reflects a health-conscious population who may be more aware of or sensitive to weight changes than the general population. As such, findings should not be generalised to all females of reproductive age, but may be most applicable to physically active or health-oriented groups. Future research should recruit more diverse samples to explore whether perceptions differ across less active populations and across different cultural contexts”.

• Abstract and Conclusion - We tempered our language to reflect this limitation. In the Abstract, we now state that the sample was predominantly UK-based and physically active. In the Conclusion, we specify that findings may be most relevant to active or health-conscious populations and highlight the need for replication in more diverse samples. The text has been amended as following: “fifteen predominantly UK-based females”

These revisions ensure that the potential sampling bias is clearly acknowledged and that our conclusions are appropriately contextualised.

3. Operationalization of “Weight Management Struggles”:

o The manuscript heavily relies on participants’ self-reported struggles with weight loss or gain without defining how "struggle" was measured or whether this referred to actual weight change or subjective difficulty.

o Recommendation: Provide clarity in the methods on how “struggle” was conceptualized and whether validated scales or anchor questions were used. Otherwise, the construct remains vague and open to interpretation. We thank the reviewer for this helpful comment. We have now clarified how “weight management struggles” were conceptualised in the study. Specifically:

• Methods section– We added the following sentence: “In this study, ‘weight management struggles’ were defined as participants’ self-reported experiences of attempting to lose weight but perceiving no change on the scales, or perceiving greater difficulty in managing weight whilst on contraceptives. This construct was subjective and designed to reflect perceptions, in line with the study’s aim to explore female experiences, rather than objectively measured weight change.”

• Limitations section– We added: “Finally, our operationalisation of ‘weight management struggles’ was based on subjective perceptions rather than objectively measured weight change, and should therefore be interpreted as reflective of perceptions rather than physiological outcomes.”

These revisions make explicit that our measure was subjective, consistent with the study’s aim to capture perceptions, while also acknowledging this as a limitation.

4. Interpretation of DMPA-Related Weight Gain:

o While DMPA is repeatedly identified as associated with weight gain, the discussion lacks nuance regarding dose, duration, and metabolic factors. The attribution is largely anecdotal and may reflect confirmation bias.

o Recommendation: Strengthen this section by discussing the limitations of self-reported weight change, the potential for recall bias, and contrasting findings from prospective studies on DMPA. Consider the role of coexisting lifestyle changes. We thank the reviewer for highlighting this valuable point. We have now expanded the Discussion section to provide greater nuance and contextualisation regarding DMPA-related weight gain, drawing on prospective evidence and acknowledging limitations of self-report. Specifically, we added the following paragraph immediately after the discussion of DMPA-related mechanisms:

“It is important to interpret these DMPA-related findings in light of methodological and contextual limitations. The current data are based on retrospective self-report, and therefore may be influenced by recall bias, particularly regarding the timing and extent of weight changes. Perceived weight gain may not always align precisely with objective measurements, and prior prospective studies provide mixed evidence. For example, Berenson and Rahman (2009) reported that over a 36-month period, DMPA users experienced significantly greater increases in body weight (+5.1 kg), fat mass (+4.1 kg), and central-to-peripheral fat ratio compared with oral and non-hormonal contraceptive users, suggesting a potential biological influence of DMPA on adiposity. In contrast, Bonny et al. (2011) examined adolescents initiating DMPA and found that early weight gain (>5% within six months) predicted continued excessive weight gain at 12–18 months. While this provides valuable insight into early predictors of weight gain, it should be noted that this study was conducted in adolescents, whose hormonal and behavioural profiles differ from adult women in the present study. Such discrepancies between studies likely reflect differences in participant age, duration of use, and individual metabolic or lifestyle factors. Overall, the current findings should therefore be viewed as reflecting a multifactorial process in which both pharmacological effects and coexisting lifestyle behaviours contribute to perceived weight change.”

This addition explicitly addresses the reviewer’s request by:

• contrasting self-reported data with prospective studies,

• discussing recall bias and methodological constraints, and

• noting that one of the cited studies focused on adolescents, distinguishing it from our adult sample.

We believe this provides the nuance and balance the reviewer requested while situating our findings within the broader evidence base.

5. Use of Social Ecological Model (SEM):

o Although SEM is introduced as the framework, its application in the findings is inconsistent. The mapping of themes onto SEM levels (e.g., interpersonal, societal) seems forced at times and lacks theoretical depth.

o Recommendation: Ensure that SEM is systematically applied across all themes with clearer justification. Avoid vague attributions (e.g., "societal level" for stress) and explain how each level influenced contraceptive perception and behavior. We appreciate this thoughtful feedback. To strengthen theoretical consistency and ensure the Social Ecological Model (SEM) is systematically applied, we have revised both the Methods and Findings and Discussion sections.

In the Methods, we now clarify how the SEM guided both coding and interpretation:

“The Social Ecological Model (SEM) was used deductively to guide coding and theme organisation, while allowing inductive sub-themes to emerge within each level. Individual-level influences captured personal beliefs, physiological experiences, and self-directed behaviours. Interpersonal factors referred to relationships and immediate social contexts (e.g., partners, peers, coaches) shaping contraceptive experiences and weight perceptions. Community and societal levels encompassed institutional or cultural influences such as workplace demands, university environments, and sociocultural body ideals. This framework allowed multi-layered interpretation of how biological, behavioural, and contextual factors interact to influence perceived weight change.”

In the Findings and Discussion, we added an integrative paragraph at the start of the section to demonstrate how the SEM was applied consistently across the analysis:

“Findings are interpreted through the Social Ecological Model to highlight how influences on perceived weight management operate across multiple, interacting layers. At the individual level, participants described physiological sensations, appetite changes, and emotional responses related to contraception. The interpersonal level encompassed support networks and social comparison, such as conversations with partners, peers, or trainers that shaped interpretations of weight gain. The community level included transitions into new environments—university, employment, or sporting settings—that altered lifestyle routines. Finally, societal influences such as cultural ideals of thinness and public discourse around contraception framed participants’ expectations and self-evaluation. Applying the SEM systematically in this way illustrates that perceptions of contraceptive-related weight change are co-constructed through personal, relational, and structural contexts rather than arising from physiology alone.”

These revisions ensure consistent and theoretically grounded application of the SEM throughout the manuscript.

Minor Comments

1. Background in Introduction:

o The introduction would benefit from a stronger contextual background on the significance of perceived weight changes in contraceptive users, especially linking this perception to contraceptive adherence and health outcomes. A more focused rationale would help set up the need for this mixed-methods study. We thank the reviewer for this valuable suggestion. We have now strengthened the Introduction to provide a clearer rationale for the study and to explicitly link perceptions of weight change with contraceptive adherence, health outcomes, and the need for a mixed-methods approach. Specifically, we added the following paragraph after the discussion of psychological and behavioural barriers to oral contraceptive use (Westhoff et al., 2007; Littlejohn, 2013):

“Perceived weight change, even in the absence of objectively measured gain, has significant implications for contraceptive adherence and broader health outcomes. Studies have shown that concerns about weight gain are among the most frequently cited reasons for disco

---

## [Decision Letter · Decision Letter 1]

14 Nov 2025

PONE-D-25-30751R1Does it hold weight? The perceived effects of contraceptive use on weight status in females: A Mixed-Methods StudyPLOS ONE?

Dear Dr. Johnson,

Thank you for submitting your manuscript to PLOS ONE. After careful consideration, we feel that it has merit but does not fully meet PLOS ONE’s publication criteria as it currently stands. Therefore, we invite you to submit a revised version of the manuscript that addresses the points raised during the review process.

We look forward to receiving your revised manuscript.

Kind regards,

Vandana Dabla

Academic Editor

PLOS ONE

Journal Requirements:

**Additional Editor Comments:**

Authors are requested to revise the manuscript as per reviewer's comments attached on the revision submitted.

In addition, it is noteworthy that while the DMPA perceptions naturally arose in interviews and thus its inclusion is vital in the study results; there is no equal discussion of individual metabolic variability, the role of lifestyle, psychosocial factors and the benefits of DMPA itself. Thus, it may cause a moderate risk of misinterpretation to reader that DMPA universally causes weight gain (which neither is evidenced by the study, nor it is its objective).

Hence, to ensure ethical neutrality and scientific fairness, the author should explicitly clarify that the "association is perceived, and is not established causally or any other biomedical evidence in the current study". Add explicit qualifiers distinguishing perceptions from evidence and may mention DMPA’s clinical safety/effectiveness in the discussion. Such ethical neutrality must be maintained throughout the manuscript, including recommendations.

Reviewers' comments:

Reviewer's Responses to Questions

**Comments to the Author**

Reviewer #1: All comments have been addressed

2. Is the manuscript technically sound, and do the data support the conclusions?

Reviewer #1: Yes

3. Has the statistical analysis been performed appropriately and rigorously?

Reviewer #1: Yes

4. Have the authors made all data underlying the findings in their manuscript fully available?

Reviewer #1: Yes

5. Is the manuscript presented in an intelligible fashion and written in standard English?

Reviewer #1: Yes

Reviewer #1: Terminology consistency: Ensure consistent use of terms like "weight management struggles," "weight gain," and "weight status." Clarify their operational definitions early in the Methods section.

Clarify measures: Explicitly state how "struggles" were assessed – subjective reports, specific questions, or scales. Emphasize the perception-based nature and acknowledge this as a limitation.

Formatting and language: Conduct a thorough proofreading to remove minor grammatical errors, ensure tense consistency, and adhere to journal style guidelines.

Title and abstract: Consider adding a subtitle for clarity, for example, "A Mixed-Methods Study." Also, clearly state the study’s location (e.g., UK) in the abstract for context.

References: Update any outdated references, remove duplicate citations, and ensure all are correctly formatted.

Ethical statement: Clarify that participants consented explicitly for quotations and data publication, emphasizing confidentiality and anonymity processes.

Additional minor points:

Expand the discussion on limitations, especially regarding the sample's representativeness and potential biases.

Elaborate briefly on the validation process of the questionnaire.

Consistently define abbreviations at first mention (e.g., DMPA).

**Do you want your identity to be public for this peer review?** For information about this choice, including consent withdrawal, please see our Privacy Policy

Reviewer #1: No

---

## [Author Response · Author response to Decision Letter 2]

3 Dec 2025

Reviewers Comments

Your Comment / Recommendation Revision

1. Authors are requested to revise the manuscript as per reviewer's comments attached on the revision submitted.

In addition, it is noteworthy that while the DMPA perceptions naturally arose in interviews and thus its inclusion is vital in the study results; there is no equal discussion of individual metabolic variability, the role of lifestyle, psychosocial factors and the benefits of DMPA itself. Thus, it may cause a moderate risk of misinterpretation to reader that DMPA universally causes weight gain (which neither is evidenced by the study, nor it is its objective).

Hence, to ensure ethical neutrality and scientific fairness, the author should explicitly clarify that the "association is perceived, and is not established causally or any other biomedical evidence in the current study". Add explicit qualifiers distinguishing perceptions from evidence and may mention DMPA’s clinical safety/effectiveness in the discussion. Such ethical neutrality must be maintained throughout the manuscript, including recommendations. We have now strengthened conceptual neutrality throughout the latter sections of the manuscript. New statements explicitly clarify that all findings reflect subjective perceptions rather than physiological evidence. We also added a statement noting that DMPA remains a highly effective and clinically safe contraceptive method. These additions appear in:

• Discussion (end of DMPA section)

• Limitations (final paragraph)

• Conclusion (final lines)

We acknowledge the need for conceptual neutrality; however, given the consistency of participant reports and the existing literature on DMPA-related weight changes, we consider it important to present these perceptions clearly while emphasising that they do not imply causality.

Terminology consistency: Ensure consistent use of terms like "weight management struggles," "weight gain," and "weight status." Clarify their operational definitions early in the Methods section. We thank the reviewer for this helpful comment. We have ensured consistent terminology throughout the manuscript and have now added a clear operational definition of all weight-related terms early in the Methods section. Specifically, we inserted a labelled paragraph defining weight management struggles, weight status, and weight gain immediately after the description of the timeline-interview procedure. These terms are now used consistently throughout the manuscript.

This has also been acknowledged in limitations.

We have also clarified how perceived weight management struggles were assess with the aim to capture subjective perceptions.

Clarify measures: Explicitly state how "struggles" were assessed – subjective reports, specific questions, or scales. Emphasize the perception-based nature and acknowledge this as a limitation.

Formatting and language: Conduct a thorough proofreading to remove minor grammatical errors, ensure tense consistency, and adhere to journal style guidelines. Grammar, tense consistency, and formatting now comply with PLOS ONE style.

Title and abstract: Consider adding a subtitle for clarity, for example, "A Mixed-Methods Study." Also, clearly state the study’s location (e.g., UK) in the abstract for context. The title already includes “a mixed-methods study”. We have highlighted this for clarity.

The abstract also already states “Three hundred and fifteen predominantly UK-based females completed a questionnaire assessing” see line 21

References: Update any outdated references, remove duplicate citations, and ensure all are correctly formatted. Completed.

Ethical statement: Clarify that participants consented explicitly for quotations and data publication, emphasizing confidentiality and anonymity processes. We have now added a clear statement in the Ethics subsection confirming that participants provided explicit consent for anonymised quotation use.

Consent explicitly included permission for anonymised quotations to be used in publications and presentations, and all identifying information was removed from transcripts to ensure participant confidentiality. Direct quotes were sent to participants for approval prior to the development on the manuscript.

Additional minor points:

Expand the discussion on limitations, especially regarding the sample's representativeness and potential biases.

Elaborate briefly on the validation process of the questionnaire.

Consistently define abbreviations at first mention (e.g., DMPA). Addressed in previous revision. The Limitations now explicitly note the UK-based, highly active sample and its implications.

The Methods now state that both content validity (expert review) and face validity (pilot testing) were applied.

DMPA comment addressed and added into the table in the introduction.

---

## [Editor Report · Decision Letter 2]

7 Dec 2025

Does it hold weight? The perceived effects of contraceptive use on weight status in females: A Mixed-Methods Study

PONE-D-25-30751R2

Dear Dr. Johnson,

We’re pleased to inform you that your manuscript has been judged scientifically suitable for publication and will be formally accepted for publication once it meets all outstanding technical requirements.

Kind regards,

Vandana Dabla

Academic Editor

PLOS One
---

## [Editor Report · Acceptance letter]

PONE-D-25-30751R2

PLOS One

Dear Dr. Johnson,

I'm pleased to inform you that your manuscript has been deemed suitable for publication in PLOS One. Congratulations! Your manuscript is now being handed over to our production team.

Kind regards,

on behalf of

Dr. Vandana Dabla

Academic Editor

PLOS One